# White Light-Emitting Flexible Displays with Quantum-Dot Film and Greenish-Blue Organic Light-Emitting Diodes

**DOI:** 10.3390/mi15121518

**Published:** 2024-12-20

**Authors:** Young Woo Kim, Seojin Kim, Chaeyeong Lee, Joo Hyun Jeong, Yun Hyeok Jeong, Yuhwa Bak, Seo Hyeon Kim, Sung Jin Park, Ko Eun Ham, Doeun Lee, Junpyo Song, Youngjin Song, Seung-Chan Jung, Oh Kwan Kwon, Jae-Hee Han, Sang Jik Kwon, Eou-Sik Cho, Yongmin Jeon

**Affiliations:** 1Department of Semiconductor Engineering, Gachon University, 1342 Seongnam-Daero, Soojung-gu, Seongnam 13120, Republic of Korea; mae04008@gachon.ac.kr (Y.W.K.); emoyo@gachon.ac.kr (S.K.); titiputi@gachon.ac.kr (C.L.); jjh042512@gachon.ac.kr (J.H.J.); dnsl0515@gachon.ac.kr (Y.H.J.); bak994@gachon.ac.kr (Y.B.); harrimiron77@gachon.ac.kr (S.H.K.); jos6251@gachon.ac.kr (S.J.P.); rhdms02@gachon.ac.kr (K.E.H.); dlehdms1998@naver.com (D.L.); junpyosong@naver.com (J.S.); anjelzin@naver.com (Y.S.); sjkwon@gachon.ac.kr (S.J.K.); 2Department of Materials Science and Engineering, Gachon University, Seongnam 13120, Republic of Korea; jsc4486@gachon.ac.kr (S.-C.J.); jhhan388@gachon.ac.kr (J.-H.H.); 3InnoQD, Co., Ltd., Dunpo-myeon, Asan-si 31412, Republic of Korea; grokkwon@innoqd.com; 4Department of Biomedical Engineering, Gachon University, 1342 Seongnam-Daero, Soojung-gu, Seongnam 13120, Republic of Korea

**Keywords:** QD-OLED, OLED, QD, white, display

## Abstract

White organic light-emitting diodes (OLEDs) represent a significant technology in the display industry for the achievement of full color. However, sophisticated technologies are required for white light emission. In this paper, we developed a simple white light-emitting display device using a quantum-dot (QD) film and a greenish-blue OLED. The resulting QD-OLED produced a high-purity white color with a color temperature of 6000 K (CIE_x,y_ = 0.32, 0.34) and achieved a maximum brightness of 14,638 cd/m^2^ at 7 V. This paper reports the fabrication of a white light-emitting QD-OLED with a straightforward structure and technology suitable for flexible displays.

## 1. Introduction

Organic light-emitting diodes (OLEDs) are devices that have been increasingly recognized in the fields of television, wearable devices, and lighting for their advantages in substrate selection and flexibility, and for their thin and lightweight characteristics compared to conventional rigid inorganic-based general light-emitting diodes (LEDs) [1,2,3,4,5]. These OLEDs are fabricated using organic materials through manufacturing processes such as inkjet printing and evaporation [6,7,8,9]. Displays and lighting applications that utilize these OLEDs can be deposited on flexible substrates of various shapes and have applications in the arts, architecture, and clothing, fostering new opportunities [9,10,11].

In this study, we attempted to fabricate a white light source using a simple method. Various technologies are employed to reproduce the color white in OLED displays. These include highly complex processes such as tandem structures [12,13,14], parallel stacking [15], and AC driving [10,16,17,18]. These stacked structures change the wavelength due to the microcavity effect. This causes problems in which certain wavelengths appear strong or disappear. To solve this, structural wavelength simulation is required. Alternatively, OLED elements are driven by AC to create a time-division white light-emitting display. This has a problem in that when the driving frequency is not high enough or when it is driven by constant power, it is essentially invisible as a white light source. Also, there are studies that use complex organic materials [19,20]. These studies reported very high driving voltages (over 10 V), but did not include information on brightness, or had a very low luminescence of 1.4 cd/m^2^ at a voltage of 5.8 V. In addition, the disadvantage were confirmed which spectrum changes depending on the voltage. Moreover, white light displays can be made with pixel miniaturization and layout technologies [21,22,23], and there are also many commercialized products. This technology requires high resolution, so it is difficult to manufacture. In addition, there is also a disadvantage in that a uniform white color does not appear when the resolution is low. Additionally, studies have reported the generation of white light by positioning green and red quantum dots (QDs) on blue emitters [24,25,26]. However, they used two colors of solution QDs, green and red, and implemented white light-emitting displays on rigid substrates, such as glass substrates. Therefore, there is no report of flexibility in these studies. To make them easier to compare, we have summarized them in Appendix A.

Furthermore, when creating a white light source, the correlated color temperature (CCT) and color rendering index (CRI) are essential metrics for assessing color accuracy [27,28]. Typically, light sources with low color temperatures (2700 to 3000 K) are categorized as warm white, those within the range of 5000 to 6000 K as pure white, and those exceeding 8000 K as cold white [10,29]. The closer the color temperature is to pure white, the more effectively the light source reproduces colors for various applications. Additionally, a high CRI enhances the light source’s ability to illuminate objects, resulting in truer colors [27,28].

A QD is a nanomaterial consisting of particles smaller than 10 nm, studied extensively as an optical semiconductor device [30]. Since the introduction of CdS-based QDs in 1983 [31], numerous studies on the topic have emerged, including those on CuInS2-based [32,33], Si-based [34,35], and more recently, graphene- and carbon-based quantum dots [36,37]. These QDs are categorized into electroluminescence (EL)-type devices, which operate by directly applying electricity, and photoluminescence (PL)-type devices, which emit light when exposed to light.

QD devices driven by the EL method exhibit a significantly higher maximum brightness compared to OLEDs, but suffer from a shorter lifespan, despite them being inorganic [38]. One study reported a lifetime of only 0.25 h in LT_50_ under 10 mW/cm^2^ irradiance [39]. By contrast, devices powered by the PL method achieve high levels of maximum emission brightness with correctly matched absorption wavelengths, and their lifespan is comparable to that of OLEDs [38].

In this paper, we introduce a white light-emitting device that can be manufactured in a flexible and easy manner, unlike in previous studies. Other QD-OLEDs have not been reported to be flexible, and white light-emitting OLEDs have poor luminescence performance or require complex technologies. This study simply discusses a method for fabricating a flexible white light source by attaching a greenish-blue OLED fabricated on an adhesive PET substrate onto a flexible QD film. Moreover, we analyzed the photoluminescence properties of QD film to achieve a higher brightness and a pure white color. Accordingly, we achieved a high brightness of up to 14,638 cd/m^2^ at an operating voltage of 7 V and a CCT of 6000 K.

## 2. Experimental Section

Blue OLED fabrication: Both an ITO-based OLED and a cavity-based OLED were deposited under a vacuum of 1.0 × 10^−7^ torr using a thermal evaporator (SELCOS Inc. Hawseong-si, Republic of Korea). The substrate of OLEDs consisted of adhesive PET with a thickness of 30 μm. The ITO-based OLED was first deposited with 150 nm of ITO using an in-line sputter with a shadow mask followed by evaporation, whereas the cavity-based OLED was deposited with 25 nm of Ag through a thermal evaporator (SELCOS Inc.) under a deposition condition of 1 Å/s, at a temperature of 950 °C. Both ITO and Ag were used as anodes, followed by the deposition of 5 nm of MoO_3_, used as an EIL, under a condition of 0.5 Å/s, at a temperature of 500 °C. Then, 45 nm of NPB, used as an ETL, was deposited under a condition of 1.0 Å/s, at a temperature of 130 °C. Next, 25 nm of MADN:DSA-ph (3 wt%), used as an EML, was co-deposited under a condition of 0.9 Å/s and at a temperature of 110 °C for MADN, and a condition of 0.1 Å/s and a temperature of 150 °C for DSA-Ph. A total of 10 nm of Bebq_2_, used as an ETL, was deposited under a condition of 1.0 Å/s, at a temperature of 120 °C. Then, 1 nm of Cs_2_CO_3_, used as an EIL was deposited under a condition of 0.1 Å/s, at a temperature of 650 °C, and 100 nm of Al, used as the cathode, was deposited under a condition of 1.0 Å/s using a boat evaporator (however, there was no temperature sensor on the boat).

QD-OLED fabrication: Each ITO-based OLED and cavity-based OLED was fabricated on an adhesive PET substrate. Therefore, the fabricated OLEDs were detached from the 150 μm guide PET and directly attached to various types of QD films. In addition, to prevent the degradation of the devices during measurement, a 30 μm thick adhesive PET was simply attached onto the OLEDs to minimize moisture and oxygen penetration into it.

Device characterization: The optical and electrical characteristics of each OLED and QD-OLED were measured using a source meter (Keithley 2601, Keithley Inc., Solon, OH, USA) and a spectrometer (CS-2000, Konica Minolta Inc., Tokyo, Japan). Additionally, we gauged the PL characteristics of the device using a PL measuring device (HORIBA Jobin Yvon NanoLog, HORIBA Instruments Inc., Kyoto, Japan).

## 3. Results

To produce natural white light emission, the lights must incorporate blue, green, and red light (RGB). Accordingly, various displays incorporate RGB into miniaturized pixels or utilize tandem structures. In this paper, we aimed to manufacture a flexible white light-emitting display using a greenish-blue OLED fabricated on a flexible substrate and a QD film emitting at 630 nm, avoiding complex manufacturing techniques, as shown in Figure 1a. The fabricated QD-OLED displays the wavelength depicted in Figure 1b. The emitted color closely approximates white light, with three peak wavelengths occurring at 465, 510, and 630 nm, corresponding to deep blue, green, and red, respectively. Additionally, the CIE 1931 coordinates were utilized to determine the displayed color of a light-emitting element. In the CIE 1931 coordinate system, the device used in this study exhibited a CCT ranging from 6000 to 6300 K, with a peak luminance at 7 V. The precise CCT was recorded as 6041 K at the position (CIE_x,y_ = 0.32, 0.34), which lies at the exact center of the CIE coordinate, qualifying it as pure white, as illustrated in Figure 1c.

For the construction of the display platform, the fabrication process is presented as illustrated in Figure 2a. Materials for the OLEDs were deposited on flexible substrates such as adhesive PET through an evaporation–deposition process. Initially, indium tin oxide (ITO) was deposited on the substrate via sputtering, followed by subsequent processing steps. Once the flexible OLED was manufactured on the adhesive PET, it could be directly attached to the QD film. The attached device emitted greenish-blue light, while the excited QDs produced red light-emission for the white light emission.

The device, upon deposition, exhibited the band structure depicted in Figure 2b. To confirm the results observed in the greenish region, a structure was employed utilizing ITO as a transparent anode and featuring a microcavity induced by thinly depositing Ag. Furthermore, the device configuration is illustrated in Figure 2c. A flexible substrate was mounted atop the QD film, onto which either 25 nm of Ag or 150 nm of ITO was deposited. Subsequently, molybdenum trioxide (MoO_3_) was deposited as a 5 nm thick hole injection layer (HIL) and N,N’-di(1-naphthyl)-N,N’-diphenyl-(1,1’-biphenyl)-4,4’-diamine (NPB) as a 45 nm hole transfer layer (HTL). Following this, 2-methyl-9,10-di(2-naphthyl)anthracene: p-bis(p-N,N-di-phenyl-aminostyryl) (MADN: DSA-Ph 3 wt%) was deposited as a 25 nm thick emission layer (EML), and bis(10-hydroxybenzo[h]quinolinato)-beryllium (Bebq_2_) was deposited as a 10 nm thick electron transfer layer (ETL). The electron injection layer (EIL) comprised 1 nm of Cs_2_CO_3_, and finally, 100 nm of Al was deposited as the cathode.

The greenish-blue light-emitting device based on ITO exhibited characteristics akin to white light emission when integrated with a QD film, whereas the deep-blue light-emitting device employing the cavity structure emitted purple light, as shown in Figure 3a,b. The performance prior to the use of each device, obtained by merging it with a QD film, can be verified in Appendix A. Maximum luminance reached 27,533 cd/m^2^ at 7 V for the ITO-based device and 27,889 cd/m^2^ at 7 V for the cavity-based device, as illustrated in Appendix A. Regarding current density, the cavity structure demonstrated superior current injection characteristics compared to ITO, as depicted in Appendix A. This improvement results from the lower anode surface resistance in the cavity structure. The full-width at half-maximum of the spectrum generated by each device was 81 nm for the ITO-based device and 22 nm for the cavity-based device, as shown in Appendix A. Additionally, while the ITO-based OLED exhibited a first peak at 465 nm and a secondary peak at 510 nm, the cavity-based OLED only displayed a peak at 465 nm. The lower brightness despite the higher current injection rate in the cavity-based devices is due to the loss of the green region of the spectrum attributed to the cavity structure.

The spectrum of each QD-OLED is depicted in Figure 3c. In comparison to the ITO-based QD-OLED, the green region of the cavity-based QD-OLED is notably reduced, confirming that purple light emission, rather than white light, is observed. The CIE 1931 color coordinates are available in Appendix A. Each CIE 1931 color coordinate correlates with the ITO-based (CIE_x,y_ = 0.32, 0.34) and the cavity-based (CIE_x,y_ = 0.42, 0.25) devices. Additionally, the luminance characteristics of devices incorporating the QD film were measured at 14,638 cd/m^2^ for the ITO QD-OLED and 15,342 cd/m^2^ for the cavity QD-OLED at a peak of 7 V, as depicted in Figure 3d. At this point, the current density displayed in Figure 3e matched that of Appendix A since the operating component remained unchanged.

For optimum white light emission, the QD film was analyzed. The structure of the QD film consists of matrix layers with varying concentrations and thicknesses between two thin substrates (<20 μm), as illustrated in Figure 4a. The QDs within the matrix layer comprise InP and are capsulated. Comparative measurements employed films with a 22.5% matrix layer concentration and QD films with a 30% concentration. PL characteristics were assessed accordingly, as shown in Figure 4b,c. In Figure 4b, the film with a 22.5% concentration displayed the highest absorption peak at 18,422 excitons in the light range of 420 nm and a significant second peak at 14,483 excitons at 465 nm. In Figure 4c, a film with a 30% concentration produced 32,818 excitons at 420 nm and a secondary peak of 27,421 excitons at 465 nm. In both instances, a PL peak was observed in the 630 nm region. At this time, the thickness of the films did not significantly influence the PL characteristics.

QD films were measured using ITO-based greenish OLEDs at various concentrations and matrix layer thicknesses. The spectrum for each film at driving voltage of 7 V is shown in Figure 4d, where the film with the best uniformity in the red region and the blue to green region had a matrix layer thickness of 100 μm at a 30% concentration, as used in previous analyses. It was also confirmed that higher concentrations transmitted more blue light. When comparing the spectral characteristics of films with a thickness of 161 μm at a 30% concentration to those with a thickness of 200 μm or more at a 22.5% concentration, it is evident that there was a minimal difference in the transmitted blue region light. This is attributed to the 22.5% concentration being the optimal capsulized concentration, enhancing light absorption.

Furthermore, in terms of brightness, the configuration with a matrix layer thickness of 100 μm at a 30% concentration achieved the highest brightness. As previously shown in Figure 3, a luminance of 14,638 cd/m^2^ was achieved, peaking at 7 V. Generally, greater thickness correlates with lower brightness, and films with similar thicknesses, high-concentration variants exhibit lower brightness than their low-concentration counterparts. In addition, the current injected into the devices was compared, revealing significant differences between the devices, with Appendix A showing similar current injections across all devices.

Lastly, when the CIE 1931 color coordinates for each film were assessed, thinner films with higher concentrations of the QD matrix layer tended to exhibit luminescence characteristics closer to white, as demonstrated in Figure 4f. This behavior results from the fact that at a 22.5% concentration, QDs absorb more light, and thinner films transmit more light in the blue to green range.

## 4. Conclusions

In summary, this paper presents a flexible QD-OLED display technology capable of white light emission through a simple method. The technique achieved pure white light emission at a color temperature of 6000 K in the optimal structure and a luminance of 14,638 cd/m^2^ at a maximum of 7 V with a CRI of 38.3.

First, we evaluated the performance and characteristics of the device using greenish-blue OLED and deep-blue OLED, both of which exhibit similar brightness characteristics and internal structures. For the greenish-blue OLED, we successfully achieved a white light source with the color coordinates (CIE_x,y_ = 0.32, 0.34). Additionally, we confirmed that the deep-blue OLED emitted purple light and compared the performance of each device. This analysis confirmed that the greenish-blue OLED is well suited for a simple flexible white light-emitting display.

However, the CRI of the optimally structured QD-OLED device fabricated in this paper had a rather poor result, ranging from 52.5 (at 4 V, 10 cd/m^2^) to 38.3 (at 7 V, 14,638 cd/m^2^). This is because the wavelength corresponding to 580 nm did not appear in the spectrum of the device. These limitations will be addressed in the future by using QDs with a wider full-width at half-maximum or by appropriately mixing QDs with an emission wavelength matching 580 nm with 630 nm QDs to form a film. It may also be possible to utilize QDs and nanorods developed for white light emission, such as those in other studies [40,41].

Through variations in the concentration and thickness of QD films, we identified films that are more effective for white light QD-OLED applications. We determined which structure best aligns with the uniformity of the red light region and the blue to green light region and established that the optimal ratio was achieved with a matrix layer thickness of 100 μm and a concentration of 30%. Furthermore, we discovered that adjusting the QD concentration or the thickness of the QD film allows for a color transition from pure white to warm white or cool white.

## Figures and Tables

**Figure 1 micromachines-15-01518-f001:**
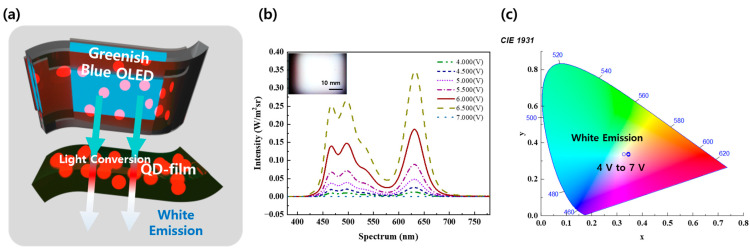
Concept and color data of QD-OLED: (**a**) schematic illustration of this work’s concept; (**b**) spectral characteristic graph against bias voltage; (**c**) CIE 1931 coordinates of QD-OLED.

**Figure 2 micromachines-15-01518-f002:**
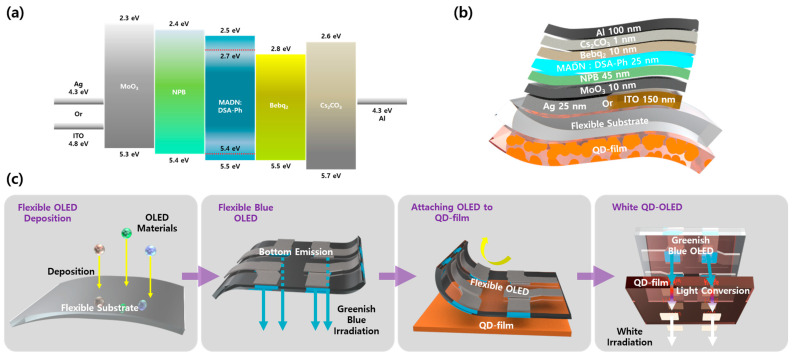
Schematic illustration of (**a**) manufacturing process of QD-OLED; (**b**) band diagram of OLED; (**c**) thickness and structure of QD-OLED.

**Figure 3 micromachines-15-01518-f003:**
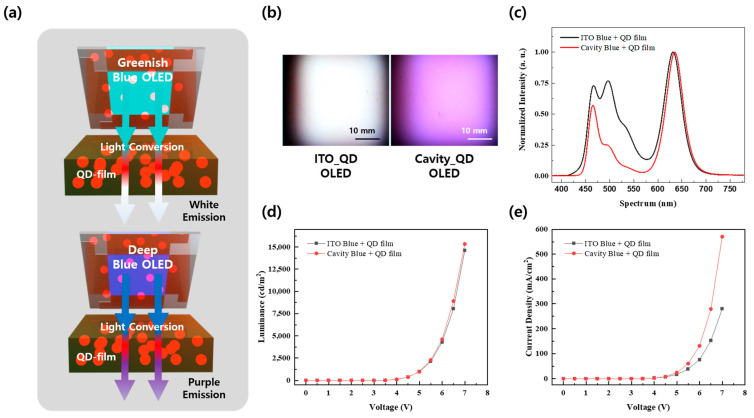
Electric and optical properties of various QD-OLEDs: (**a**) schematic illustration of QD-OLEDs; (**b**) emission image of QD-OLEDs; (**c**) spectral characteristics of QD-OLEDs; (**d**) voltage versus luminance graph of QD-OLEDs, (**e**) voltage versus current density graph of QD-OLEDs.

**Figure 4 micromachines-15-01518-f004:**
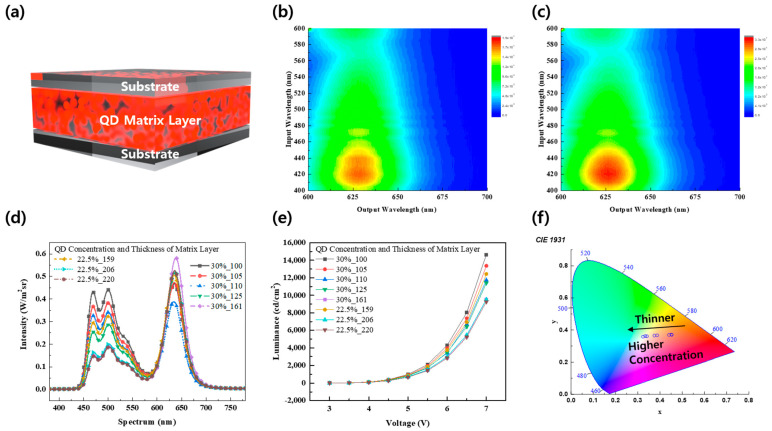
Optical properties of various QD films: (**a**) schematic illustration of a QD-film; (**b**) PL characteristics of 22.5% concentration QD film; (**c**) PL characteristics of 30% concentration QD film; (**d**) spectral characteristics of various QD-OLEDs against the concentration and thickness of the QD film; (**e**) voltage vs. luminance graph of various QD-OLEDs; (**f**) CIE 1931 coordinates of various QD-OLEDs.

## Data Availability

The data that support the findings of this study are available from the corresponding author upon reasonable request.

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
