# Peer review of "White Light-Emitting Flexible Displays with Quantum-Dot Film and Greenish-Blue Organic Light-Emitting Diodes"

_micromachines, 2024, doi:10.3390/mi15121518_

Round 1

Reviewer 1 Report

Comments and Suggestions for Authors

The manuscript "White Emitting Flexible Displays with Quantum-Dots Film and Greenish Blue Organic Light Emitting Diodes" presents a significant advancement in display technology. The authors successfully developed a straightforward method for fabricating white-emitting QD-OLEDs, achieving impressive brightness levels of 14,638 cd/m² and a color temperature of 6000K. Their approach combines the flexibility of OLEDs with the photoluminescent properties of quantum dots, making it suitable for various applications, including wearable devices.

However, some improvements are suggested.

One suggestion regarding the change in one statement line: “These OLEDs are fabricated using organic materials and manufacturing processes such as inkjet printing and evaporation processes [6-9]." Suggested Revision: "These OLEDs are fabricated using organic materials through manufacturing processes such as inkjet printing and evaporation."

The manuscript lacks sufficient detail in the fabrication process. The authors should provide step-by-step protocols, including specific conditions (e.g., temperature, time) for each deposition step, to enhance reproducibility.

I recommend including a comparative table that benchmarks the performance of the proposed QD-OLED against existing technologies, highlighting key metrics such as luminance, efficiency, and lifespan, to clearly demonstrate the advancements your research offers over prior work.

Avoid redundancy in phrases. For example, "the device configuration is schematically illustrated" can be shortened to "the device configuration is illustrated."

When presenting numerical data, ensure clarity by specifying units consistently (e.g., "cd/m²" instead of "cd/m2" or “10-7 torr”) and using appropriate formatting. 

Correct typographical errors such as "schmatic" instead of "schematic" in figure captions

Author Response

Responses to the Reviewer 1’s comments

Reviewer 1) : The manuscript "White Emitting Flexible Displays with Quantum-Dots Film and Greenish Blue Organic Light Emitting Diodes" presents a significant advancement in display technology. The authors successfully developed a straightforward method for fabricating white-emitting QD-OLEDs, achieving impressive brightness levels of 14,638 cd/m² and a color temperature of 6000K. Their approach combines the flexibility of OLEDs with the photoluminescent properties of quantum dots, making it suitable for various applications, including wearable devices.

However, some improvements are suggested.

Response: We appreciate reviewers commnet. We have applied the improvements through manuscript revisions based on the reviewer's comments.

Reivewer 1, Comment 1) : One suggestion regarding the change in one statement line: “These OLEDs are fabricated using organic materials and manufacturing processes such as inkjet printing and evaporation processes [6-9]." Suggested Revision: "These OLEDs are fabricated using organic materials through manufacturing processes such as inkjet printing and evaporation.

Response: We appreciate reviewers commnet about the changing the statement for the clear understand. We understand that the sentence “These OLEDs are fabricated using organic materials and manufacturing processes such as inkjet printing and evaporation processes [6-9]." were not good explanation.

In addition, we’ve changed the statement line by reviewers recommendation as follows:

These OLEDs are fabricated using organic materials through manufacturing processes such as inkjet printing and evaporation

On page #1 in the revised manuscript.

Reivewer 1, Comment 2) : The manuscript lacks sufficient detail in the fabrication process. The authors should provide step-by-step protocols, including specific conditions (e.g., temperature, time) for each deposition step, to enhance reproducibility.

Response: We appreciate reviewers commnet about the needs of improving the details of fabrication process. As your opinion by this review, for the clear understand we clearified and add some additional fabricational process about this research.

In addition, the following content was added to the revised manuscript to mention fabrication process.

Blue OLED fabrication: Both ITO-based OLED and Cavity OLED were deposited under a vacuum of 1.0 ⅹ 10-7 torr using a thermal evaporator (SELCOS Inc.). The substrate of OLEDs consists of adhesive PET with a thickness of 30 μm. The ITO-based OLED was first deposited with 150 nm of ITO using an in-line sputter with the shadow mask followed by evaporation, whereas the Cavity-based OLED was deposited with 25 nm of Ag through thermal evaporator (SELCOS Inc.) with the deposition condition of 1 Å/s, temperature of 950 °C. Both ITO and Ag were used as anodes, followed by the deposition of 5 nm of MoO3 as EIL was deposited with the condition of 0.5 Å/s, temperature of 500 °C. 45 nm of NPB as ETL was deposited with the condition of 1.0 Å/s, temperature of 130 °C. 25 nm of MADN:DSA-ph (3 wt%) as EML was co-deposited with the condition of 0.9 Å/s, temperature of 110 °C for MADN, and 0.1 Å/s, temperature of 150 °C for DSA-Ph. 10 nm of Bebq2 as ETL was deposited with the condition of 1.0 Å/s, temperature of 120 °C. 1 nm of Cs2CO3 as EIL was deposited with the condition of 0.1 Å/s, temperature of 650 °C., and 100 nm of Al as the cathode was deposited with the condition of 1.0 Å/s used boat (however, there is no temperature sensor on the boat).

QD-OLED fabrication: Each ITO-based OLED and Cavity OLED was fabricated on an adhesive PET substrate. Therefore, the fabricated OLEDs were detached from the 150 μm guide PET and directly attached to various types of QD-films. In addition, to prevent degradation of the devices during measurement, a 30 μm thick adhesive PET was simply attached onto the OLEDs to minimize moisture and oxygen penetration to it.

On page #7 in the revised manuscript.

Reivewer 1, Comment 3) : I recommend including a comparative table that benchmarks the performance of the proposed QD-OLED against existing technologies, highlighting key metrics such as luminance, efficiency, and lifespan, to clearly demonstrate the advancements your research offers over prior work.

Response: We appreciate reviewers commnet about the needs of comparing the specific performances with other devices.

In this study, we attempted to fabricate a white light source using a simple method. Other studies use a tandem structure or an AC drive structure that stacks multiple elements in a single pixel. These stacked structures change wavelength due to the microcavity effect. This causes problems in which certain wavelengths appear strong or disappear. To solve this, structural wavelength simulation is required. Alternatively, OLED elements are driven by AC to create a time-division white display. This has the problem that when the driving frequency is not high enough, or when driven by constant power it is essentially invisible as a white light source.

In addition, many research and commercialized products implement white color by miniaturizing pixels and arranging them structurally. This technology requires high resolution, so it is difficult to manufacture. In addition, there is also a disadvantage in that it does not appear as a uniform white color when the resolution is low.

There is a technology to make the organic material itself emit white light. The studies reported in the text have very high driving voltages (over 10 V), do not include information on brightness, or have very low luminescence of 1.4 cd/m2 at a voltage of 5.8 V. In addition, the disadvantage of spectrum change depending on the voltage is confirmed.

Similar to our research, there are also reports on papers that produced white light sources using QDs and blue OLEDs. These studies also successfully implemented white color, but they used two colors of solution QDs, green and red, and implemented white displays on rigid substrates, such as glass substrates. There is no report on flexibility in these studies.

However, in our study, we were able to obtain a flexible white light source in a simple way, and easily solved the technical complexity, cost issues, and low luminescence scale listed above.

We hope this table will help to explain that our device has novel way to fabricate wihte emission devices, and also clearly demonstrate the advancements of our research.

In addition, the following content was added to the revised supporting information and revised manuscripts to demonstrate advancement of our research.

Table S1. Comparison between several different white emission devices

Devices

Maximum luminance

Color temperature

Flexibility

Method for white emission

Reference

QD-OLED

14,638 cd/m2
@ 7 V

6041 K

OLED attach to QD-film

This work

OLED

N/A

3000 K

×

AC driving

[10]

OLED

N/A

4860 K

×

Tandem

[12]

OLED

15,000 cd/m2
@
N/A

3000 K

×

Tandem

[13]

OLED

10,200 cd/m2
@ 40 V

4660 K

×

Tandem

[14]

OLED

30,000 cd/m2
@ 8.5 V

N/A

Parallel-Stack
(Tandem)

[15]

OLED

5,219 cd/m2

@ 15 V

4500 K

×

Organic material engineering

[19]

OLED

400 cd/m2

@ > 15 V

5200 K

×

Organic material engineering

[20]

OLED

N/A

6500 K

×

Pixel miniaturization

[22]

OLED

4,000 cd/m2

@ 20 V

5100 K

×

Organic material engineering

[23]

QD-OLED

964 cd/m2
@ 8 V

2574 K

×

Spin coated QD solution thin film

[24]

QD-OLED

4133 cd/m2

@ 12 V

6200 K

×

Spin coated QD solution thin film

[25]

Perovskite QD-OLED

49,000 cd/m2

@ 12 V

5600 K

×

Spin coated QD solution thin film

[26]

On page #2 in the revised supporting information.

In addition, the following content was added to the revised manuscripts for improved explanation.

In this study, we attempted to fabricate a white light source using a simple method. Various technologies are employed to reproduce the color white in OLED displays. These include highly complex processes such as tandem structures [12-14], parallel stacking [15], AC driving [10, 16-18]. These stacked structures change wavelength due to the microcavity effect. This causes problems in which certain wavelengths appear strong or disappear. To solve this, structural wavelength simulation is required. Alternatively, OLED elements are driven by AC to create a time-division white display. This has the problem that when the driving frequency is not high enough, or when driven by constant power it is essentially invisible as a white light source. Also, there is studies that use complex organic materials[19, 20]. These studies reported very high driving voltages (over 10 V), do not include information on brightness, or have very low luminescence of 1.4 cd/m2 at a voltage of 5.8 V. In addition, the disadvantage of spectrum change depending on the voltage is confirmed. In addition, white displays can be made with pixel miniaturization and layout technologies [21-23], and there are also many commercialized products. This technology requires high resolution, so it is difficult to manufacture. In addition, there is also a disadvantage in that it does not appear as a uniform white color when the resolution is low. Additionally, studies have reported generating white light by positioning green and red quantum dots (QDs) on blue emitters [24-26]. However, they used two colors of solution QDs which are green and red, and implemented white displays on rigid substrates, such as glass substrates. Therefore there is no report on flexibility in these studies. To make them easier to compare, we've summarized them in supporting information Table S1.

On page #1-2 in the revised manuscript.

Reivewer 1, Comment 4) : Avoid redundancy in phrases. For example, "the device configuration is schematically illustrated" can be shortened to "the device configuration is illustrated.".

Response: We appreciate reviewers commnet. We have reflected the corrections to duplicate phrases.We made revisions to the parts suggested by the reviewer, and we also wanted to fix other parts that contained repetitive explanations, but fortunately we couldn't find any.

In addition, the following content was fixed to the revised manuscripts.

[Furthermore, the device configuration is schematically illustrated in Figure 2 (c).] was changed to [Furthermore, the device configuration is illustrated in Figure 2 (c).

On page #3 in the revised manuscript.

Reivewer 1, Comment 5) : When presenting numerical data, ensure clarity by specifying units consistently (e.g., "cd/m²" instead of "cd/m2" or “10-7 torr”) and using appropriate formatting.

Response: Thank reviewer for pointing out the errors in the units of data that occurred intermittently. We have corrected the errors in the superscript of the square expression and the errors in the part written about pressure that the reviewer pointed out. Also, we checked for errors in superscripts, subscripts, spacing, etc. and corrected them.

In addtion the following contents were correted in the reviesd manuscript.

The QDs within the matrix layer comprise InP and are capsulated. Comparative measurements employed films with a 22.5 % matrix layer concentration and QDfilms with a 30 % concentration. PL characteristics were assessed accordingly, shown in Figures 4 (b, c). For Figure 4 (b), the film with a 22.5 % concentration displayed the highest absorption peak at 18,422 excitons in the light range of 420 nm and a significant second peak at 14,483 excitons at 465 nm. For Figure 4 (c), a film with a 30 % concentration produced 32,818 excitons at 420 nm and a secondary peak of 27,421 excitons at 465 nm. In both instances, a PL peak was observed in the 630 nm region.

film with the best uniformity in the red region and the blue to green region had a matrix layer thickness of 100 μm at a 30 % concentration, as used in previous analyses. It was also confirmed that higher concentrations transmitted more blue light. When comparing the spectral characteristics of films with a thickness of 161 μm at a 30 % concentration to those with a thickness of 200 μm or more at a 22.5 % concentration, it is evident that there is minimal difference in the transmitted blue region light. This is attributed to the 22.5 % concentration being the optimal capsulized concentration, enhancing light absorption.

Furthermore, in terms of brightness, the configuration with a matrix layer thickness of 100 μm at a 30 % concentration achieved the highest brightness. As previously reported in Figure 3, a luminance of 14638 cd/m2 was achieved, peaking at 7 V. Generally, greater thickness correlates with lower brightness, and films with similar thicknesses demonstrate that high-concentration variants exhibit lower brightness than their low-concentration counterparts. In addition, the injected current into the devices was compared, revealing significant differences between the devices, with Figure S3 showing similar current injections across all devices.

Lastly, when assessing the CIE 1931 color coordinates for each film, Figure 4 (f) demonstrates that thinner films with higher concentrations of the QD matrix layer tend to exhibit luminescence characteristics closer to white. This behavior results from the fact that at a 22.5 % concentration, QDs absorb more light, and thinner films transmit more light in the blue to green range.

On page #5-6 in the revised manuscript.

Blue OLED fabrication: Both ITO-based OLED and Cavity OLED were deposited under a vacuum of 1.0 ⅹ 10-7 torr using an thermal evaporator (SELCOS Inc.).

On page #7 in the revised manuscript.

Reivewer 1, Comment 6) : Correct typographical errors such as "schmatic" instead of "schematic" in figure captions

Response: We appreciate reviewers commnet about the needs of correcting typographical error. We have corrected all errors in the figure captions based on the reviewer's comments.

In addtion the following contents were correted in the reviesd manuscript.

Figure 1. Concept and color measure data of QD-OLED: (a) schematic illustration of this work’s concept, (b) spectral characteristic graph against bias voltage, (c) CIE 1931 coordination of QD-OLED.

On page #3 in the revised manuscript.

Figure 2. Schematic illustration of: (a) manufacturing process on QD-OLED, (b) band diagram of OLED, (c) thickness and structure of QD-OLED.

On page #4 in the revised manuscript.

Figure 3. Electric and optical properties of various QD-OLEDs: (a) schematic illustration of QD-OLEDs, (b) emission image of QD-OLEDs, (c) spectral characteristics of QD-OLEDs, (d) voltage versus luminance graph of QD-OLEDs, (e) voltage versus current density graph of QD-OLEDs.

On page #5 in the revised manuscript.

Figure 4. Optical properties of various QD-films: (a) schematic illustration of a QD-film, (b) PL characteristic of 22.5 % concentration QD-film, (c) PL characteristic of 30 % concentration QD-film, (d) spectral characteristics of various QD-OLEDs against concentration and thickness of QD-film, (e) voltage vs luminance graph of various QD-OLEDs, (f) CIE 1931 coordination of various QD-OLEDs.

On page #6 in the revised manuscript.

Reviewer 2 Report

Comments and Suggestions for Authors

In order to facilitate the preparation of white-light emitting display, a simple white light-emitting display device is developed by combining the flexibility of organic light-emitting diodesOLEDsand the photoluminescence characteristics of quantum-dots (QD) films. The performance and characteristics of greenish blue OLED and deep blue OLED are evaluated, and it is concluded that greenish blue OLED is very suitable for simple flexible white-emitting display. In addition, through changes in the concentration and thickness of the QD films, a more effective film for white QD-OLED applications was identified. However, the following modifications are proposed to further enhance the manuscript:

1、     Clearly mention the novelty of the study in the final paragraph of the Introduction section.

2、     In addition to the brightness and the correlated color temperature, it is necessary to provide the color rendering index of the prepared QD-OLED.

3、     There is a lack of comparison between the performance of the prepared white QD-OLED and the existing literature data. Please add information about other literature data to the main manuscript for reference.

4、     Please unify the orientation of the axes of the figures in the article. For example, Graph c, d and e in Figure 3, the coordinate scale of Graph c is outward, and the coordinates of the other two graphs are inward.

5、     It is recommended to change the line types in some figures to improve the readability, such as Figure 4d.

6、     Please consider adding the following references in the manuscript: 

Chemical Engineering Journal, 2023, 471, 144578

Materials Today Nano, 2024, 25, 100457

Advanced Optical Materials, 2024, 12, 2301427

Journal of Colloid and Interface Science, 2020, 567, 235

Small, 2021, 17, 2007397.

Comments on the Quality of English Language

It is OK.

Author Response

Responses to the Reviewer 2’s comments

Reviewer 2) : In order to facilitate the preparation of white-light emitting display, a simple white light-emitting display device is developed by combining the flexibility of organic light-emitting diodesOLEDsand the photoluminescence characteristics of quantum-dots (QD) films. The performance and characteristics of greenish blue OLED and deep blue OLED are evaluated, and it is concluded that greenish blue OLED is very suitable for simple flexible white-emitting display. In addition, through changes in the concentration and thickness of the QD films, a more effective film for white QD-OLED applications was identified. However, the following modifications are proposed to further enhance the manuscript:

Response: We would like to thank the reviewers for their positive comments on this paper. We have discussed the improvements you have pointed out below to improve the quality of this study.

Reivewer 2, Comment 1) :  Clearly mention the novelty of the study in the final paragraph of the Introduction section.

Response: We appreciate reviewers commnet.We discussed the novelty of this paper at the end of the introduction according to the reviewer's opinion. In addition, we discussed the differences from other studies in the middle of the introduction and prepared a supplementary table.

The following contents are added to the end of the introduction.

In this paper, we introduce a white light-emitting device that can be manufactured in a flexible and easy way, unlike previous studies. Other QD-OLEDs have not been reported to be flexible, and white-emitting OLEDs have poor luminescence performance or require complex technologies. This study simply discusses a method to fabricate a flexible white light source by attaching a greenish blue OLED fabricated on an adhesive PET substrate onto a flexible QD-film. Moreover, we analyzed the photoluminescence properties of QD-film to achieve higher brightness and pure white color. Accordingly, we have achieved a high brightness of up to 14638 cd/m2 at an operating voltage of 7 V and a CCT of 6000 K.

On page #2 in the revised manuscript.

Also, to support the argument in the last part of the introduction, we have added the content below.

Table S1. Comparison between several different white emission devices

Devices

Maximum luminance

Color temperature

Flexibility

Method for white emission

Reference

QD-OLED

14,638 cd/m2
@ 7 V

6041 K

OLED attach to QD-film

This work

OLED

N/A

3000 K

×

AC driving

[10]

OLED

N/A

4860 K

×

Tandem

[12]

OLED

15,000 cd/m2
@
N/A

3000 K

×

Tandem

[13]

OLED

10,200 cd/m2
@ 40 V

4660 K

×

Tandem

[14]

OLED

30,000 cd/m2
@ 8.5 V

N/A

Parallel-Stack
(Tandem)

[15]

OLED

5,219 cd/m2

@ 15 V

4500 K

×

Organic material engineering

[19]

OLED

400 cd/m2

@ > 15 V

5200 K

×

Organic material engineering

[20]

OLED

N/A

6500 K

×

Pixel miniaturization

[22]

OLED

4,000 cd/m2

@ 20 V

5100 K

×

Organic material engineering

[23]

QD-OLED

964 cd/m2
@ 8 V

2574 K

×

Spin coated QD solution thin film

[24]

QD-OLED

4133 cd/m2

@ 12 V

6200 K

×

Spin coated QD solution thin film

[25]

Perovskite QD-OLED

49,000 cd/m2

@ 12 V

5600 K

×

Spin coated QD solution thin film

[26]

On page #2 in the revised supporting information.

In this study, we attempted to fabricate a white light source using a simple method. Various technologies are employed to reproduce the color white in OLED displays. These include highly complex processes such as tandem structures [12-14], parallel stacking [15], AC driving [10, 16-18]. These stacked structures change wavelength due to the microcavity effect. This causes problems in which certain wavelengths appear strong or disappear. To solve this, structural wavelength simulation is required. Alternatively, OLED elements are driven by AC to create a time-division white display. This has the problem that when the driving frequency is not high enough, or when driven by constant power it is essentially invisible as a white light source. Also, there is studies that use complex organic materials[19, 20]. These studies reported very high driving voltages (over 10 V), do not include information on brightness, or had very low luminescence of 1.4 cd/m2 at a voltage of 5.8 V. In addition, the disadvantage of spectrum change depending on the voltage is confirmed. In addition, white displays can be made with pixel miniaturization and layout technologies [21-23], and there are also many commercialized products. This technology requires high resolution, so it is difficult to manufacture. In addition, there is also a disadvantage in that it does not appear as a uniform white color when the resolution is low. Additionally, studies have reported generating white light by positioning green and red quantum dots (QDs) on blue emitters [24-26]. However, they used two colors of solution QDs which are green and red, and implemented white displays on rigid substrates, such as glass substrates. Therefore there is no report on flexibility in these studies. To make them easier to compare, we've summarized them in supporting information Table S1.

On page #1-2 in the revised manuscript.

Reivewer 2, Comment 2) : In addition to the brightness and the correlated color temperature, it is necessary to provide the color rendering index of the prepared QD-OLED.

Response: We appreciate reviewers commnet about the needs of color rendering index (CRI) of these device. The CRI of the QD-OLEDs used in this paper are as follows. The optimally structured QD-OLED (30% concentration and 100 μm matrix layer conditions) shows a CRI value ranging from a maximum of 52.5 (at 4 V, 10 cd/m2) to 38.3 (at 7 V, 14,638 cd/m2). Other conditions record lower CRI coefficients. The reason why the CRI is low while providing a color temperature close to pure white is because the spectrum corresponding to 580 nm is not displayed. This is a limitation of this study, but it is a problem that can be solved by using a QD film with a wider FWHM.

This part is included in the conclusion sections of this paper as follows:

In summary, this paper presents a flexible QD-OLED display technology capable of white emission through a simple method. The technique achieved pure white emission at a color temperature of 6000 K in the optimal structure and a luminance of 14638 cd/m2 at a maximum of 7 V with the CRI of 38.3.

On page #6 in the revised manuscript.

However, the CRI of the optimally structured QD-OLED device fabricated in this paper was a rather poor result, ranging from 52.5 (at 4 V, 10 cd/m2) to 38.3 (at 7 V, 14,638 cd/m2). This is because the wavelength corresponding to 580 nm does not appear in the spectrum of the device. These limitations will be addressed in the future by using QDs with a wider full width half maximum or by appropriately mixing QDs with an emission wavelength matching 580 nm with 630 nm QDs to form a film.It may also be possible to utilize QDs and nanorods developed for white light emission, such as those in the following studies. [40,41].

On page #6-7 in the revised manuscript.

Reivewer 2, Comment 3) : There is a lack of comparison between the performance of the prepared white QD-OLED and the existing literature data. Please add information about other literature data to the main manuscript for reference.

Response: We appreciate reviewers commnet about the needs of comparing existing literature data. As we answered in the previous comment 1, we added comparative content to the introduction and added supplementary table 1 accordingly.

In addition, to support the argument in the last part of the introduction, we have added the content below.

Table S1. Comparison between several different white emission devices

Devices

Maximum luminance

Color temperature

Flexibility

Method for white emission

Reference

QD-OLED

14,638 cd/m2
@ 7 V

6041 K

OLED attach to QD-film

This work

OLED

N/A

3000 K

×

AC driving

[10]

OLED

N/A

4860 K

×

Tandem

[12]

OLED

15,000 cd/m2
@
N/A

3000 K

×

Tandem

[13]

OLED

10,200 cd/m2
@ 40 V

4660 K

×

Tandem

[14]

OLED

30,000 cd/m2
@ 8.5 V

N/A

Parallel-Stack
(Tandem)

[15]

OLED

5,219 cd/m2

@ 15 V

4500 K

×

Organic material engineering

[19]

OLED

400 cd/m2

@ > 15 V

5200 K

×

Organic material engineering

[20]

OLED

N/A

6500 K

×

Pixel miniaturization

[22]

OLED

4,000 cd/m2

@ 20 V

5100 K

×

Organic material engineering

[23]

QD-OLED

964 cd/m2
@ 8 V

2574 K

×

Spin coated QD solution thin film

[24]

QD-OLED

4133 cd/m2

@ 12 V

6200 K

×

Spin coated QD solution thin film

[25]

Perovskite QD-OLED

49,000 cd/m2

@ 12 V

5600 K

×

Spin coated QD solution thin film

[26]

On page #2 in the revised supporting information.

In this study, we attempted to fabricate a white light source using a simple method. Various technologies are employed to reproduce the color white in OLED displays. These include highly complex processes such as tandem structures [12-14], parallel stacking [15], AC driving [10, 16-18]. These stacked structures change wavelength due to the microcavity effect. This causes problems in which certain wavelengths appear strong or disappear. To solve this, structural wavelength simulation is required. Alternatively, OLED elements are driven by AC to create a time-division white display. This has the problem that when the driving frequency is not high enough, or when driven by constant power it is essentially invisible as a white light source. Also, there is studies that use complex organic materials[19, 20]. These studies reported very high driving voltages (over 10 V), do not include information on brightness, or had very low luminescence of 1.4 cd/m2 at a voltage of 5.8 V. In addition, the disadvantage of spectrum change depending on the voltage is confirmed. In addition, white displays can be made with pixel miniaturization and layout technologies [21-23], and there are also many commercialized products. This technology requires high resolution, so it is difficult to manufacture. In addition, there is also a disadvantage in that it does not appear as a uniform white color when the resolution is low. Additionally, studies have reported generating white light by positioning green and red quantum dots (QDs) on blue emitters [24-26]. However, they used two colors of solution QDs which are green and red, and implemented white displays on rigid substrates, such as glass substrates. Therefore there is no report on flexibility in these studies. To make them easier to compare, we've summarized them in supporting information Table S1.

On page #1-2 in the revised manuscript.

Reivewer 2, Comment 4) :  Please unify the orientation of the axes of the figures in the article. For example, Graph c, d and e in Figure 3, the coordinate scale of Graph c is outward, and the coordinates of the other two graphs are inward.

Response: We appreciate reviewers comment that pointing out the axes and errors of the figures used in the paper. The relevant part was modified as follows:

On page #3 in the revised manuscript.

On page #5 in the revised manuscript.

On page #6 in the revised manuscript.

On page #3 in the revised supporting information.

On page #4 in the revised supporting information.

Reivewer 2, Comment 5) :  It is recommended to change the line types in some figures to improve the readability, such as Figure 4d.

Response: We appriciate reviewer for the comment. Based on your comment, we have modified the figure 4d in the paper to include symbols to ensure visibility.

In addition the following contents are fixed on the revised manuscripts.

On page #6 in the revised manuscript.

Reivewer 1, Comment 6) : Please consider adding the following references in the manuscript: 

Chemical Engineering Journal, 2023, 471, 144578

Materials Today Nano, 2024, 25, 100457

Advanced Optical Materials, 2024, 12, 2301427

Journal of Colloid and Interface Science, 2020, 567, 235

Small, 2021, 17, 2007397.

Response: We would like to thank the reviewers for suggesting excellent papers. Among the papers suggested by the reviewers, [Materials Today Nano, 2024, 25, 100457], and [Journal of Colloid and Interface Science, 2020, 567, 235] were judged to be very good for improving the quality of this paper. Therefore, these papers were discussed in the conclusion and added to the references.

The following contents and references are added to the revised manuscripts.

However, the CRI of the optimally structured QD-OLED device fabricated in this paper was a rather poor result, ranging from 52.5 (at 4 V, 10 cd/m2) to 38.3 (at 7 V, 14,638 cd/m2). This is because the wavelength corresponding to 580 nm does not appear in the spectrum of the device. These limitations will be addressed in the future by using QDs with a wider full width half maximum or by appropriately mixing QDs with an emission wavelength matching 580 nm with 630 nm QDs to form a film.It may also be possible to utilize QDs and nanorods developed for white light emission, such as those in the following studies. [40,41].

On page #6-7 in the revised manuscript.

  1. Xue, Q.; Cai, P.; Pu, X.; Ai, Q.; Si, J.; Yao, X.; Bai, G.; Dong, Q.; Liu, Z. Green synthesis of high-quality indium phosphide quantum dots using tripyrrolidine phosphine as a promising phosphorus source for white LED. Materials Today Nano 2024, 25, 100457, doi:https://doi.org/10.1016/j.mtnano.2024.100457.
  2. Sun, P.; Wang, Z.; Sun, D.; Bai, H.; Zhu, Z.; Bi, Y.; Zhao, T.; Xin, X. pH-guided self-assembly of silver nanoclusters with aggregation-induced emission for rewritable fluorescent platform and white light emitting diode application. Journal of Colloid and Interface Science 2020, 567, 235-242, doi:https://doi.org/10.1016/j.jcis.2020.02.016.

On page #10 in the revised manuscript.
